# The Epigenetic Regulation of HCC Metastasis

**DOI:** 10.3390/ijms19123978

**Published:** 2018-12-10

**Authors:** Tae-Su Han, Hyun Seung Ban, Keun Hur, Hyun-Soo Cho

**Affiliations:** 1Korea Research Institute of Bioscience and Biotechnology, Daejeon 34141, Korea; tshan@kribb.re.kr (T.-S.H.); banhs@kribb.re.kr (H.S.B.); 2Department of Biochemistry and Cell Biology, School of Medicine, Kyungpook National University, Daegu 41944, Korea

**Keywords:** epigenetics, miRNA, DNA methylation, histone methylation, HCC, metastasis

## Abstract

Epigenetic alterations, such as histone modification, DNA methylation, and miRNA-mediated processes, are critically associated with various mechanisms of proliferation and metastasis in several types of cancer. To overcome the side effects and limited effectiveness of drugs for cancer treatment, there is a continuous need for the identification of more effective drug targets and the execution of mechanism of action (MOA) studies. Recently, epigenetic modifiers have been recognized as important therapeutic targets for hepatocellular carcinoma (HCC) based on their reported abilities to suppress HCC metastasis and proliferation in both in vivo and in vitro studies. Therefore, here, we introduce epigenetic modifiers and alterations related to HCC metastasis and proliferation, and their molecular mechanisms in HCC metastasis. The existing data suggest that the study of epigenetic modifiers is important for the development of specific inhibitors and diagnostic targets for HCC treatment.

## 1. Introduction

Hepatocellular carcinoma (HCC), a common primary liver cancer, is the second leading cause of death in cancer patients, and is caused by chronic hepatitis B and C virus (HBV and HCV) infections and other factors, such as alcohol, diabetes, and aflatoxin exposure [1,2]. In addition to surgical removal, chemotherapy using sorafenib, a multikinase inhibitor, is recognized as an effective treatment for HCC. Treatment with sorafenib, in both in vivo and in vitro studies, suppressed cell proliferation, metastasis, and angiogenesis via inactivation of VEGFRs, Raf-1, and B-Raf [3]. However, sorafenib resistance in advanced HCC is considered a serious problem that must be overcome for HCC therapy. Therefore, the identification of new therapeutic targets and the development of specific inhibitors for HCC is required [4]. Recently, several papers reported that various epigenetic modifiers and alterations (e.g., microRNA (miRNA), histone modification, and DNA methylation) participate in HCC proliferation and metastasis via epigenetic regulation of oncogenes and tumor suppressor genes, suggesting that epigenetic modifiers are important therapeutic targets for the development of specific inhibitors of HCC [5,6,7].

Thus, in this review, we introduce HCC-related epigenetic modifiers (miRNA, DNA methyltransferases/demethylases, and histone methyltransferases/demethylases) and suggest their clinical utility in HCC treatment. In particular, we summarize the molecular mechanisms and functions of epigenetic modifiers in HCC metastasis, suggesting their potential as therapeutic targets and diagnostic markers in HCC.

## 2. MicroRNA

miRNAs regulate gene expression at the posttranscriptional level by inhibiting the translation or decay of transcripts, depending on the target sequence matching ratio. The biogenesis of miRNAs has been extensively studied. In brief, primary miRNAs (pri-miRNAs) are generated by RNA polymerase II and processed into precursor miRNAs (pre-miRNAs) by the RNase III enzyme Drosha and the DiGeorge critical region 8 (DGCR8) microprocessor complex in nucleus [8,9]. Subsequently, pre-miRNAs are exported to the cytoplasm by exportin 5 [10], where they are processed by Dicer and TAR RNA-binding protein (TRBP) into 21–24 nucleotide miRNA duplexes that are unwound and incorporated into the RNA-induced silencing complex (RISC) to mature miRNA [11,12]. However, defects in miRNA biogenesis machinery are frequently observed in various cancers [13,14,15]. Furthermore, abnormal regulation of miRNA expression is linked to tumor progression and metastasis [16,17]. In this miRNA section, we will discuss epithelial–mesenchymal transition (EMT)-related miRNAs in HCC and their oncogenic or tumor suppressive functions (Table 1).

### 2.1. EMT-Related miRNAs

#### 2.1.1. TGF-β Signaling Pathway-Related miRNAs

Transforming growth factor (TGF)-β signaling has been shown to play an important role in EMT. TGF-β induces cells to lose their epithelial characteristics and to acquire migratory behavior through activating Smad (mothers against decapentaplegic homolog) signaling [18]. The major downstream genes of TGF-β signaling are Smad2/3 and Smad4, which act as transcription factors to change the expression of EMT-related genes, including those in the Snail and zinc-finger E-box-binding homeobox (ZEB) families [19,20].

Recent studies have indicated that abnormal regulation of miRNA-mediated TGF-β/Smad signaling pathways induce malignant tumor development. Low expression of miR-542-3p and miR-142 is frequently found in HCC; these two miRNAs directly regulate the *TGFB1* transcript by binding to the 3’-UTR [21,22] and, therefore, negatively regulate the TGF-β/Smad signaling pathway, implicating them in EMT and metastasis in HCC. However, it has been reported that Smad7, an inhibitor of TGF-β signaling, is also regulated by miRNAs. High expression of miR-216a/217 promoted EMT, cell migration, stem-like properties, and HCC recurrence by targeting the Smad7, and phosphatase and tensin homolog (PTEN) tumor suppressor genes [23].

Several studies have shown that miRNAs regulate various downstream genes of TGF-β signaling, but this does not occur through Smad signaling. For example, TGF-β1-induced EMT was suppressed by miR-300 through targeting focal adhesion kinase (FAK), which modulates EMT [24]. Another study found an inverse correlation between miR-199b-5p and the expression of its target gene, N-cadherin, in HCC. Restoration experiments showed that miR-199-5p overexpression in HCC suppressed TGF-β1-induced Akt phosphorylation, and inhibition of the PI3K/Akt pathway blocked TGF-β1-induced N-cadherin overexpression [25].

Some studies have reported that miRNAs can be positively or negatively regulated by TGF-β signaling. For example, treatment with recombinant TGF-β1 increased miR-155 and miR-181a expression levels in HCC [26,27]. Overexpression of these miRNAs by TGF-β signaling led to increased EMT, with similar phenotypes observed after TGF-β treatment in vitro, by blocking E-cadherin (CDH1), and promoting Snail and ZEB1 in HCC.

#### 2.1.2. WNT Signaling Pathway-Related miRNAs

There are several lines of evidence that indicate that the Wnt/β-catenin signaling pathway plays important roles in EMT [46]. In particular, the nuclear localization of β-catenin increases the expression of target genes, such as fibronectin and matrix metalloproteinase-7 (MMP-7); consequently, cells adopted mesenchymal-like phenotypes [47].

Wang et al. found that the downregulation of miR-122 enhanced the proliferation, migration, and invasion of HCC. However, they showed that miR-122 overexpression decreased cell proliferation, migration, and invasion by targeting Wnt1, and inhibiting EMT-related gene expression [29]. miR-148a also has a binding site in the Wnt1 3’-UTR, and negatively regulates Wnt1 expression in HCC [30]. Therefore, the expression of miR-122 and miR-148a attenuates the Wnt signaling pathway by inhibiting Wnt1 expression, thereby suppressing progression through EMT and cancer stem cell (CSC)-like properties in HCC.

The Wnt/β-catenin signaling pathway can regulate miRNA expression levels. For example, miR-25 is significantly upregulated in human HCC tissues compared with normal liver tissues. The functions of miR-25 include stimulating HCC cell growth and activating EMT by targeting Rho GDP dissociation inhibitor alpha (RhoGDI1) [31]. miR-25 is upregulated by the Wnt/β-catenin signaling pathway. miR-25 expression levels are positively correlated with β-catenin levels, and negatively correlated with RHOGDI1 levels in HCC.

#### 2.1.3. Snail-, Slug-, and Twist1-Related miRNAs

Snail1 is the critical point of convergence in EMT regulation and represses CDH1 expression at the transcriptional level [48]. A previous report showed that miR-1306-3p activation by Forkhead box protein M1 (FOXM1) directly inhibits the F-box/LRR-repeat protein 5 (FBXL5) gene, which regulates Snail protein stability by binding to its 3’-UTR. Upregulation of miR-1306-3p by FOXM1 suppresses FBXL5, resulting in increased Snail stability. Therefore, high expression of miR-1306-3p promoted EMT progression [35]. miR-122 downregulation is associated with tumor invasion and metastasis in HCC. Functional studies have shown that the upregulation of miR-122 inhibits cell proliferation and EMT by targeting the expression of Snail1 and Snail2 [36]. miR-30a and miR-30b have been reported to negatively regulate Snail1 in HCC, but the downregulation of miR-30a and miR-30b in HCC affects EMT by increasing Snail expression levels [37,38]. Furthermore, Snail1 is also negatively regulated by miR-153 in HCC. Ectopic expression of miR-153 inhibited migration, invasion, and EMT, and decreased EMT marker levels [39].

Previously, miR-140-5p and miR-630 were found to directly bind to Slug and negatively regulate its expression in HCC [40,28]. Another study showed that miRNAs upregulated by KLF4 included miR-153, miR-506, and miR-200b, which target Snail1, Slug, and ZEB1 mRNA, respectively [41].

Recent reports have shown that Twist1 induces EMT and promotes metastasis in HCC because it regulates various EMT-associated genes. Significant downregulation of miR-26b-5p and miR-27a-3p has been found in HCC. Mechanistically, Twist1 could suppress these miRNAs by binding to their promoter regions [32,33,34]. Previous studies showed that overexpression of miR-26b-5p and miR-27a-3p suppressed EMT and invasion by targeting Smad1 and VE-cadherin, respectively.

In addition, high expression of miR-345 inhibited EMT and cell mobility by targeting IRF1-mediated mTOR/STAT3/AKT signaling, and genes downstream of these pathways, including Snail, Slug, and Twist, are related to EMT in HCC [42].

### 2.2. Metastasis-Related Exosomal miRNAs

Exosomes, a type of extracellular vesicle, are small vesicles less than 200 nm in diameter [49]. In the human body, exosomes secreted by cells exist in body fluids, such as serum and urine, and contain DNA, mRNA, noncoding RNA, and protein. Exosomes move to other cells and communicate via their contents.

A previous study showed that a high level of miR-103 was associated with a higher metastasis potential of HCC. Exosomal miR-103 increases vascular permeability and promotes metastasis by directly targeting endothelial junction proteins, including VE-cadherin (VE-Cad), p120-catenin (p120), and zonula occludens 1 (ZO1). Therefore, miR-103 is a therapeutic target and metastasis marker of HCC [43]. Another study showed that highly metastatic liver cancer cells secreted exosomal miR-1247-3p, which affected cancer-associated fibroblasts (CAFs). The functions of miR-1247-3p include suppressing B4GALT3 expression and promoting lung metastasis [44]. A previous report showed that CAF-derived miR-320a is related to HCC progression and metastasis. A low level of exosomal miR-320a of CAF promotes HCC progression and metastasis by targeting the PBX3-Erk1/2 pathway in HCC [45]. Thus, exosomal miRNAs might be used not only as diagnostic or prognostic markers, but also as therapeutic targets for the treatment of malignant HCC.

## 3. DNA Methylation

DNA methylation catalyzed by DNA methyltransferases (DNMTs) is a chemical modification of DNA that occurs by conjugation of a methyl group to the 5’ carbon position of the cytosine ring, which is crucial for regulating gene expression [50]. In cancer, the expression of tumor suppressor genes is frequently silenced by CpG island hypermethylation in promoter regions [51]. Therefore, the upregulation of DNMTs promotes cancer development [52]. For example, the mRNA levels of DNMTs, including DNMT1, DNMT3a, and DNMT3b, are significantly higher in HCC than in nonneoplastic liver tissues [53]. In addition, the increased expression of DNMTs is recognized as a predictor of poor survival in HCC [54]. To date, numerous studies have demonstrated that DNMT-mediated epigenetic changes regulate HCC metastasis, invasion, progression, and development [55]. This section provides information on the regulation of HCC metastasis by DNMTs and their molecular mechanism (Table 2).

### 3.1. DNMT1

To date, several studies on the roles of DNMT1 in HCC metastasis have been reported. In CD133+/CD44+ cells, a subpopulation of HCC with CSC properties, a noncollagenous bone matrix protein osteopontin (OPN), enhances HCC metastasis by regulating DNA methylation. Knockdown of OPN in CD133+/CD44+ cells suppressed sphere formation and migration by inhibiting DNMT1 expression, which reduced the methylation of tumor suppressor genes such as RASSF1, GATA4, and CDKL2 [56].

The axis governed by hepatocyte growth factor (HGF) and its receptor c-Met plays important roles in cell proliferation, survival, and migration in the liver [65]. Within the HCC microenvironment, epigenetic upregulation of c-Met is associated with tumor progression and metastasis [66]. Epigenetic analysis of the c-Met promoter clarified that a significant reduction in DNA methylation is correlated with increased c-Met expression in circulating tumor cells (CTCs) during hematogenous metastasis of HCC [66]. Another study on the role of HGF in HCC metastasis reported that the induction of DNMT1 expression by HGF resulted in DNA hypermethylation of tumor suppressor genes, including MYOCD, PANX2, and LHX9 [57]. It has been reported that reactive oxygen species (ROS) are involved in tumor metastasis, migration, invasion, and tumor angiogenesis [67]. In HCC cells, ROS inhibit the expression of E-cadherin, a regulator of EMT, by inducing epigenetic changes in the promoter [58]. Moreover, the regulatory mechanism was clarified as follows: ROS upregulate Snail expression by activating the PI3K/Akt/GSK3β pathway and, then, Snail induces CpG methylation of the E-cadherin promoter by recruiting DNMT1 [58].

### 3.2. DNMT3

Several recent studies have shown that DNMT3 regulates HCC invasion and metastasis. A clinicopathological study by Oh et al. reported significant correlations between DNMT expression and overall survival and metastasis-free survival in HCC patients [59]. Among the DNMTs, DNMT3b showed a greater than 4-fold increase in mRNA levels in HCC compared to nonneoplastic livers, and these increased levels were associated with poorer overall survival and a shorter metastasis-free survival interval [59].

In HCC metastasis and invasion, DNMT3 is involved in the epigenetic regulation of the metastasis-associated protein 1 (MTA1) gene [68]. MTA1 is known as a modulator of cancer-promoting processes, including invasion, angiogenesis, metastasis, and survival [69]. In HBV-associated HCC, the HBV X (HBx) protein enhances MTA1 expression by epigenetic regulation. A molecular mechanism has been clarified in which HBx induces MTA1 transcription by increasing promoter methylation and releasing p53 through the recruitment of DNMT3a and DNMT3b [60]. In addition to gene silencing by DNMT3b through epigenetic modification, a DNA methylation-independent mechanism in HCC metastasis has also been identified. A study by Fan et al. demonstrated that metastasis suppressor 1 (MTSS1) is a novel target gene of DNMT3b, and its expression was negatively associated with DNMT3b overexpression in clinical HCC specimens [61]. The mechanism of action (MOA) by which DNMT3b inhibits MTSS1 expression was determined to involve direct binding to the MTSS1 promoter region rather than DNA hypermethylation [61].

### 3.3. Undefined DNMTs

To date, several studies on the epigenetic regulation of tumor suppressor gene expression by undefined DNMTs have been reported in HCC. PCDH10 was recently demonstrated to be a tumor suppressor gene, and is frequently silenced in HCC [70]. An analysis of the clinicopathologic features of HCC patients found a significant correlation between the methylation status of PCDH10 and metastasis [62]. Treatment with the DNA methyltransferase inhibitor 5-aza-2’-deoxycytidine (Aza) restored PCDH10 mRNA expression by inhibiting promoter methylation, indicating that decreased PCDH10 expression is related to the promoter methylation status in HCC [62].

The transmembrane glycoprotein CD147 has been implicated in HCC progression and metastasis, and CD147 gene silencing reduced MMP secretion and the invasive potential of HCC cells [71]. Within HCC patient tissues, an inverse correlation between CD147 expression and methylation status has been reported, and a mechanism study with Aza revealed that promoter hypomethylation enhances CD147 expression by increasing Sp1 binding [63]. In addition, SLIT2 has been proposed to be a tumor suppressor gene that is epigenetically silenced in various cancers [72,73,74]. In HCC, a reduction in SLIT2 expression by promoter methylation correlated with lymph node metastasis [64]. Moreover, overexpression of SLIT2 in the SMMC-7721 HCC cell line reduced cell growth, invasion, and migration [64].

Therefore, DNA methylation status and DNMT levels may be potential biomarkers of HCC and attractive therapeutic targets for HCC treatment.

## 4. Histone Modifications

Histone modifications, such as histone methylation, acetylation, and ubiquitination, critically control oncogenes and tumor suppressor genes at the transcriptional level during tumor progression [75,76]. Among the histone modifying enzymes, histone methyltransferases and demethylases are the primary focus in studies on molecular targets for anticancer drug development [77]. In HCC, several recent papers have reported that histone methylation is tightly connected to the up- and downregulation of metastasis- and proliferation-related genes. To initiate HCC metastasis from primary tumors, various methyltransferases and demethylases methylate histone H3 lysine (K) 4 and K36, which causes conformational changes affecting the balance and distribution of euchromatin and heterochromatin, leading to the upregulation of mesenchymal–epithelial transition (MET)-related genes. In contrast, these genes are involved in the downregulation of EMT-related genes after the formation of heterochromatin through methylation of histone H3K9 and K27 [78]. Therefore, here, we introduce the overexpression of histone methyltransferases and demethylases in HCC, and discuss the role of these histone modifying enzymes in HCC metastasis and proliferation (Table 3).

### 4.1. Histone Lysine Methyltransferase

Enhancer of zeste homolog 2 (EZH2) is a member of the polycomb group complex and plays an important role in the proliferation and metastasis of various cancers via methylation of histone H3K27 [92]. EZH2 mRNA is highly expressed in HCC cell lines and primary HCC tumors; moreover, clinicopathological analysis of EZH2 revealed a statistically significant difference in portal vein invasion between the high expression group and the low expression group [93]. Regarding the epigenetic regulation of tumor suppressor genes, the upregulation of chromodomain helicase DNA binding protein 5 (CHD5), which acts as a tumor suppressor in several types of cancer [94], by shEZH2, led to a reduction in cell migration and invasion, and a negative correlation between EZH2 and CHD5 was observed in HCC patients [79]. Additionally, the upregulation of tumor suppressor deleted in liver cancer 1 (DLC1) and four and a half LIM domains 1 (FHL1) by DZNep, an EZH2-specific inhibitor, also suppressed the proliferation and migration of HCC cell lines [80,81]. Recently, several papers have reported on the regulation of EZH2 and miRNA in HCC metastasis. miR-173 suppressed migration and invasion by downregulating the EZH2-STAT3 signaling pathway [95], and the upregulation of miR-22 and miR-203 by epigenetic regulation of EZH2 decreased HCC metastasis [82,83]. Moreover, miR-124 and miR-26a decreased EZH2 levels to reduce HCC metastasis in vivo and in vitro [96,97].

SETDB1 (KMT1E) is a methyltransferase that targets histone H3K9 methylation to repress gene expression [98]. SETDB1 knockdown and overexpression studies, in vivo and in HCC cell lines, clearly established its involvement in cell proliferation. In addition, downregulation of T-lymphoma invasion and metastasis gene (Tiam1), by SETDB1 knockdown in HCC cell lines, reduced EMT and cell migration/invasion. Finally, a positive correlation between SETDB1 and Tiam1 was observed in HCC tissues [84]. The transcription factor SP1 is involved in the positive regulation of SETDB1 in HCC, and overexpression of miR-29a significantly suppressed SETDB1 expression [85].

Euchromatic histone lysine methyltransferase 2 (G9a, EHMT2) and suppressor of variegation 3-9 homolog 1 (SUV39H1) predominantly methylate histone H3K9 to induce the formation of heterochromatin, and are overexpressed in several types of cancer [99,100,101]. In HCC, G9a upregulation was significantly associated with aggressive clinicopathological features of HCC, and G9a knockdown suppressed HCC cell metastasis and proliferation via the induction of retinoic acid receptor responder protein 3 (RARRES3) [86]. In addition, suppression of G9a expression by treatment with basil polysaccharide under hypoxia also induced EMT markers and reduced MET markers [102]. Upregulation of SUV39H1 in HCC cell lines markedly enhanced cell clonogenicity. In addition, SUV39H1 knockdown by the overexpression of miR-125b, a posttranscriptional regulator of SUV39H1, induced cell senescence, and reduced cell migration and metastasis [87].

### 4.2. Histone Lysine Demethylases

KDM5C and JARID1B are histone demethylases in the family of JmjC domain-containing proteins that mainly demethylate histone H3K4 to suppress gene expression via the formation of heterochromatin, and are overexpressed in many types of cancer [103,104]. In HCC, KDM5C and JARID1B are abundantly expressed in invasive human HCC cells, and are correlated with distant metastasis in HCC. The roles of KDM5C and JARID1B in HCC have been clarified by various experiments—for example, knockdown of KDM5C and JARID1B reduced migration, invasion, and wound healing via epigenetic silencing of BMP7 and PTEN expression, respectively [88,89]. Demethylation of histone H3K9 and K27 leads to the formation of open chromatin structures; knockdown of KDM4B, a H3K9 demethylase, decreased miR-615-5p expression in HCC cell lines. Subsequently, a reduction in miR-615-5p expression induced HCC growth and metastasis, in vitro, through increased RAB24 expression [90]. The expression of KDM6B (H3K27 demethylase) is critically regulated by miR-941 levels in HCC cells. Low KDM6B levels in response to miR-941 regulation reduced cell proliferation, migration, and invasion in vitro and in vivo [91]

## 5. Clinical Application of Epigenetic Alterations as Hepatic Metastasis Biomarkers and Epigenetic Modifiers as Therapeutic Targets

Cancer metastasis, the spread of cancer cells from the primary site, is the major cause of morbidity and mortality in various cancers. HCC metastasis is defined as either intrahepatic metastasis (IHM) via portal vein dissemination, or extrahepatic metastasis (EHM) to other organs, including the lungs, lymph nodes, bones, and adrenal glands [105,106,107]. Although IHM is the most effective predictor of the risk of recurrence after curative resection [108], approximately 80% of EHM patients present at a more advanced intrahepatic tumor stage at diagnosis [106,109], and EHM has also been detected in patients with early stage intrahepatic HCC [110]. Therefore, early detection of metastasis is crucial to make adequate treatment plans and to avoid unnecessary therapy. Recently, advanced imaging techniques have improved the detection rate of small tumors and the characterization of hepatic lesions. Nonetheless, there are still limits on the prediction and early diagnosis of hepatic metastases.

EMT and MET have been proposed to be engines of metastasis in various cancers, including HCC [111,112,113]. During EMT, E-cadherin is repressed, and β-catenin undergoes nuclear translocation; these events are significantly correlated with IHM and poor survival of HCC patients. Similarly, E-cadherin is strongly expressed in nonmetastatic HCC tissue, whereas it is weakly expressed in metastatic tissues [114]. In analyses of EMT-related markers’ expression using the 123 HCC tissue cohort, overexpression of Snail and Twist was correlated with the downregulation of E-cadherin expression, and was related to a worse disease prognosis [115]. Another study of EMT-related genes in a cohort of 128 HCC patients revealed that four candidate genes (E-cadherin (*CDH1*), inhibitor of DNA binding 2 (*ID2*), matrix metalloproteinase 9 (*MMP9*), and transcription factor 3 (*TCF3*)) were independent prognostic factors for the overall survival of HCC patients [116]. Moreover, low E-cadherin and high vimentin expression were significantly correlated with poor tumor differentiation, vascular invasion, and extrahepatic recurrence in 150 HCC patients [117]. Intriguingly, EMT also plays critical roles during liver cell fibroproliferative wound healing, which can lead to fibrosis, and is associated with chronic inflammation and repair of adult tissues after injury. Several previous studies demonstrated that human hepatocytes and cholangiocytes can be induced to undergo EMT by TGF-β or hedgehog, and they actively contribute to liver cell fibrogenesis [118,119,120].

Since EMT is a crucial event in hepatocyte progression and metastasis, epigenetic alterations have potential as clinically applicable biomarkers and therapeutic targets in HCC. (Figure 1) As we described above, several transcription factors, such as SNAIL, SLUG, ZEB1, and ZEB2/SIP1, participate in the repression of epithelial markers (E-cadherin) and the conversion of epithelial-like cancer cells into metastatic cancer cells. In this context, multiple aspects of epigenetic regulation, including DNA methylation, histone modification, and noncoding RNAs, are cooperatively engaged in EMT regulation. The inverse correlation between E-cadherin expression and hypermethylation of the *CDH1* gene promoter has been confirmed in many cancers [121,122,123]. In addition, increased methylation of CDH1 was related to worse overall survival, which was associated with vascular invasion and recurrence in HCC [122,123]. Notably, aberrant cytosine methylation itself was reported as a possible molecular marker for HCC progression, and treatment with the DNMT inhibitor decitabine inhibited the invasiveness of HCC cells [124,125]. Several EMT transcriptional factors also epigenetically regulate *CDH1* gene expression. Snail recruits the histone demethylase lysine-specific demethylase 1 (LSD1), which removes the dimethylation of K4 on histone H3 (H2K4m2) and mediates the transcriptional repression of CDH1 [126]. Elevated Snail levels predict the poor survival of HCC patients [127].

Currently, noncoding RNAs have been highlighted as new regulators of various genes, including EMT-related genes. The best-known EMT-related miRNAs are those in the miR-200 family (miR-200a, miR-200b, miR-200c, miR-141, and miR-429) and miR-205 [113,128]. Expression of genes in the miR-200 family is reversibly controlled by promoter CpG methylation during EMT and MET in cancer metastasis. In addition, Bundh et al. discovered 20 HCC metastasis-specific miRNAs via miRNA microarray analysis of 29 metastatic HCC tissues vs. 102 nonmetastatic HCC tissues [129]. This 20-miRNA signature could significantly predict HCC metastasis status with an overall accuracy of 76%. In further analysis, miR-219-5p was found to be an important metastasis-related miRNA in HCC [130]. CDH1 was directly regulated by miR-219-5p, and high miR-219-5p expression in HCC tissues was associated with vascular invasion and poor prognosis in HCC patients.

In terms of cancer biomarkers, the body fluid-based liquid biopsy concept has recently emerged for the noninvasive analysis of biomarkers. Indeed, many research groups have discovered and published on circulating HCC-derived biomolecules. Circulating cell-free methylated DNA from cancer cells has been detected in HCC patient body fluids, such as serum, plasma, and urine. Hypermethylated DNA from the Ras association domain family protein 1A (*RASSF1A*) gene was detectable in over 90% of HCC patient sera samples, and predicted a shorter relapse-free survival interval for HCC patients [131]. Interestingly, another group demonstrated that the methylation status of 5 tumor suppressor genes (*APC*, *FHIT*, *p15*, *p16*, and *E-cadherin*) in HCC tissues was successfully reproduced in plasma circulating DNA [132]. Thus, circulating methylated DNA is readily detectable and has potent clinical utility, such as in early HCC diagnosis and prognosis prediction. Nonetheless, HCC metastasis-specific circulating DNA methylation has not yet been reported.

Circulating miRNAs are another potential noninvasive marker for HCC liquid biopsy. A nested case-control study with prospectively collected sera from HCC patients and controls revealed significantly increased serum levels of miR-29a, miR-29c, miR-133a, miR-143, miR-145, miR-192, and miR-505 in patients with HCC [133]. In addition, Ding et al. reported the high frequency expression of 3 miRNAs (miR-21, miR-199, and miR-122) based on a meta-analysis, and these miRNAs may be more specific for the diagnosis of HCC [134]. Moreover, multiple circulating miRNAs have been reported as candidate markers for the detection of HCC. HCC patients with HBV or HCV infection had relatively high circulating levels of miR-21, miR-222, and miR-223 [135,136]. Serum miRNA-34 was proposed as a predictive marker of bone metastasis in patients with HCC [137].

In recent years, exosomes, a type of extracellular microvesicle, have been shown to be vehicles for miRNAs. Therefore, exosomal miRNAs may be stable in body fluids due to the protective role of exosomes against RNase. Exosomes carrying cancer-specific miRNAs are released from cancer cells and can circulate in body fluids. After being delivered into acceptor cells, exosomal miRNAs play functional roles by targeting specific genes. miR-21 plays an oncogenic role in various cancer tissues. Recently, high levels of exosomal miR-21 were reported in serum from HCC patients, and were shown to be significantly associated with HCC patient prognosis [138]. Notably, miR-21 was highly concentrated in exosomes compared to whole serum, which indicates that exosomal miR-21 is a more sensitive marker. Other exosomal miRNAs have also been suggested as diagnostic markers of HCC. For instance, the levels of exosomal miR-18a, miR-221, miR-222, and miR-224 were significantly higher in the serum of patients in the HCC group than in the hepatitis and liver cirrhosis groups [139].

Collectively, EMT plays critical role in HCC development and metastasis. EMT is modulated via several epigenetic changes, such as DNA methylation, histone modifications, miRNAs, indicating each epigenetic modifier has great potential to serve as a novel diagnostic and/or therapeutic marker for HCC metastasis. Thus, future research should be more focused on the development of specific inhibitors, discovery of cancer metastasis biomarker, and MOA studies based on epigenetic modifiers.

## Figures and Tables

**Figure 1 ijms-19-03978-f001:**
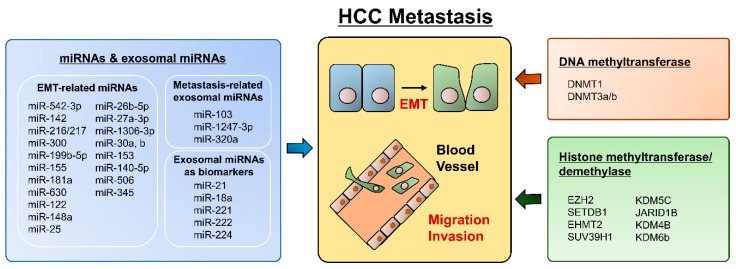
Summary for epigenetic regulation in HCC metastasis.

**Table 1 ijms-19-03978-t001:** Epithelial–mesenchymal transition (EMT)-related miRNAs and their target genes in hepatocellular carcinoma (HCC).

Signaling Pathway	miRNA	Regulation	Expression	Target Genes	Functions	References
TGF-β	miR-542-3p	-	Down	TGFB1	Proliferation, EMT	[21]
	miR-142	DNA methylation	Down	TGFB1	Proliferation, EMT	[22]
	miR-216/217	-	Up	Smad7, PTEN	Migration, EMT	[23]
	miR-300	-	Down	FAK	Invasion, EMT	[24]
	miR-199b-5p	-	Down	N-cadherin	EMT, Metastasis	[25]
	miR-155	-	Up	E-cadherin	Invasion, EMT	[26]
	miR-181a	TGF-β	Up	BIM	EMT	[27]
	miR-630	TGF-β	Down	Slug	EMT	[28]
WNT	miR-122	-	Down	Wnt1	Migration, Invasion	[29]
	miR-148a	-	Down	Wnt1	Migration, EMT	[30]
	miR-25	-	Up	RhoGDI1	Proliferation, EMT	[31]
Twist1	miR-26b-5p	Twist1	Down	Smad1	Invasion, EMT	[32]
	miR-27a-3p	Twist1	Down	VE-cadherin	EMT	[33,34]
Snail	miR-1306-3p	FOXM1	Up	FBXL5	EMT, Metastasis	[35]
	miR-122	-	Down	Snail1, Snail2	Proliferation, EMT	[36]
	miR-30a, b	-	Down	Snail1	EMT, Metastasis	[37,38]
	miR-153	-	Down	Snail1	Invasion, EMT	[39]
Slug	miR-140-5p	HBV/Unigene56159	Down	Slug	EMT	[40]
	miR-506	KLF4	Down	Slug	EMT	[41]
Other	miR-345	-	Down	IRF1	EMT	[42]
Exosomal miRNAs	miR-103	-	Up	VE-Cad, p120, ZO1	Migration, Metastasis	[43]
	miR-1247-3p	-	Up	B4GALT3	Metastasis	[44]
	miR-320a	-	Down in CAFs	PBX3	Proliferation, Metastasis	[45]

**Table 2 ijms-19-03978-t002:** DNA methyltransferases in HCC metastasis.

Genes	Function	References
DNMT1	Osteopontin induces HCC metastasis through increasing DNMT1 expression and the hypermethylation of RASSF1, GATA4, and CDKL2	[56]
	HGF-mediated HCC metastasis is associated with the induction of DNMT1 expression and the hypermethylation of MYOCD, PANX2, and LHX9	[57]
	ROS induces E-cadherin promoter methylation through Snail-dependent DNMT1 recruitment	[58]
DNMT3	Increased DNMT3b expression in HCC patients is associated with poorer overall survival and a shorter metastasis-free survival interval	[59]
	HBx induces DNMT3a and 3b recruitment and MTA1 promoter hypermethylation, which interferes with the DNA binding of p53	[60]
	DNMT3b reduces the expression of the metastasis suppressor MTSS1 via a DNA methylation-independent mechanism	[61]
Undefined DNMTs	Inhibition of DNMT by Aza restores downregulated PCDH10 expression in HCC	[62]
	Promoter hypomethylation upregulates CD147 expression by increasing Sp1 binding	[63]
	The reduction in SLIT2 expression by promoter methylation correlates with lymph node metastasis in HCC	[64]

**Table 3 ijms-19-03978-t003:** Histone methyltransferase/demethylase in HCC metastasis.

Function	Reference
**Histone methyltransferase**	
Upregulation of CHD5, DLC1, FHL1, miR-22, and miR-203 by EZH2 knockdown-reduced HCC metastasis	[79,80,81,82,83]
Downregulation of Tiam1 and SP1 by SETDB1-decreased EMT and cell migration/invasion	[84,85]
Induction of RARRES3 by EHMT2 knockdown-suppressed HCC metastasis and proliferation	[86]
Knockdown of SUV39H1 by miR-125b-induced cell senescence and metastasis	[87]
**Histone demethylase**	
Epigenetic silence of BMP7 and PTEN by KDM5C and JARID1B knockdown-reduced cell invasion/migration and wound healing analysis	[88,89]
miR-615-5p downregulation knocked down KDM4B levels, which increased RAB24 expression and induced HCC growth and metastasis	[90]
miR-941 decreased KDM6B levels to reduce cell migration and invasion in vitro and in vivo	[91]

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
