# Peer review of "The Epigenetic Regulation of HCC Metastasis"

_ijms, 2018, doi:10.3390/ijms19123978_

Round 1
Reviewer 1 Report
The review article entitled “The Epigenetic Regulation of HCC Metastasis” by Han et al. is written on the fundamentals and topics about epigenetic alterations of HCC. This manuscript is compact, but carries exhaustive information about epigenetic alterations related to HCC metastasis. Besides it, the descriptions on this paper are correctly based on the previous papers. I would think that this article is easy to understand for most of readers. would think that this article is easy to understand for most of readers.
Author Response
We thank the Reviewer for accepting our Manuscript.
Reviewer 2 Report
This review summarized the current knowledge about the epigenetic regulation of HCC metastasis. Before it can be accepted for publication, some edits are needed as listed below.
1. Table 1. "Regulation" column, if no comment, please show (-) (or something like this) for clarity.
2. Table 1. "Target genes" VM is "vasculogenic mimicry" ?
3. Line 88, Ref [28] is not suitable for reference in this context, because [28] is a paper that is specific to prostate cancer.
4. Figure 1. Please highlight especially important targets among these miRNAs etc. (" What is especially important among these miRNAs etc.? ")
in terms of HCC metastasis, not simply show the collection of known miRNA etc.
Author Response
Reviewer 2’s comment:
This review summarized the current knowledge about the epigenetic regulation of HCC metastasis. Before it can be accepted for publication,
some edits are needed as listed below.
1. Table 1. "Regulation" column, if no comment, please show (-) (or something like this) for clarity.
2. Table 1. "Target genes" VM is "vasculogenic mimicry" ?
Response:
The Authors thank the Reviewer for helpful and detailed guidance on improving our manuscript. We filled in the blanks with (-) in the Table 1.
We are sorry for our mistake that was incorrect target gene in the Table 1 and Line 124. The “VM” term was deleted from Table 1 and Line 124 both, and we add correct target gene, “VE-cadherin”.
Table 1, we changed the term of “miR-26-5p” to “miR-26b-5p”.
Table 1, we revised the incorrect references for miR-27a-3p.
3. Line 88, Ref [28] is not suitable for reference in this context, because [28] is a paper that is specific to prostate cancer.
Response:
The Authors are grateful for the guidance provided by Reviewer. We found suitable reference for this context. Here is the changed reference:
Zhang, Q.; Bai, X.; Chen, W.; Ma, T.; Hu, Q.; Liang, C.; Xie, S.; Chen, C.; Hu, L.; Xu, S.; Liang, T., Wnt/beta-catenin signaling enhances hypoxia-induced epithelial-mesenchymal transition in hepatocellular carcinoma via crosstalk with hif-1alpha signaling. Carcinogenesis 2013, 34, (5), 962-73.
4. Figure 1. Please highlight especially important targets among these miRNAs etc. (" What is especially important among these miRNAs etc.? ")
in terms of HCC metastasis, not simply show the collection of known miRNA etc.
Response:
The Authors thank the Reviewer for comments aimed to improve our manuscript. To find and highlight important miRNAs in HCC metastasis, the authors checked our suggesting miRNAs. However, we could not decide important target miRNAs, because all miRNAs have important and different roles for HCC metastasis. Therefore, in response to the Reviewer’s comment, we try to categorize the miRNAs as three groups, which are EMT-related miRNAs, Metastasis-related exosomal miRNAs and Exosomal miRNAs as biomarkers, to improve readability.